# High School Physical Education Teachers’ Perceptions of Blended Learning One Year after the Onset of the COVID-19 Pandemic

**DOI:** 10.3390/ijerph182111146

**Published:** 2021-10-24

**Authors:** Iván López-Fernández, Rafael Burgueño, Francisco Javier Gil-Espinosa

**Affiliations:** 1Department of Languages, Arts and Sports, University of Malaga, 29010 Malaga, Spain; javiergil@uma.es; 2Comprehensive and Lifelong Physical Education (CALPE) Research Group, 29010 Malaga, Spain; rafael.burgueno@ui1.es or; 3Department of Physical Education, University Isabel I, 09003 Burgos, Spain

**Keywords:** COVID-19, physical education, sport pedagogy, physical activity, hybrid education, online teaching, secondary school, adolescents

## Abstract

The COVID-19 pandemic has altered the educational landscape worldwide. One year after the disease outbreak, blended learning, which combines distance and face-to-face learning, became an alternative to fully online learning to address the demands of ensuring students’ health and education. Physical education teachers faced an additional challenge, given the experiential nature of their subject, but research on teachers’ perspectives is scarce. This study aims to explore high school physical education teachers’ perceptions of the potential, advantages, and disadvantages of the blended learning model of instruction. An online survey was used to register the views of 174 Spanish high school physical education teachers (120 men and 54 women). The main findings revealed that physical education teachers considered that blended learning, compared with full face-to-face learning, implied a work overload, worsened social relationships, and did not help to increase students’ motivation. Likewise, most teachers considered the physical activity performed by students during the blended learning period as being lower than usual. Furthermore, teachers reported that the students from lower-income families were the ones that experienced a lack of technological means the most. These results may guide both present and future policies and procedures for blended physical education. More research is needed to analyze the usefulness of blended learning in high school physical education.

## 1. Introduction

The emergence of the COVID-19 pandemic has impacted education, with most countries around the world temporarily closing educational institutions in an attempt to contain the spread of the pandemic. In January 2021, one year into the COVID-19 pandemic, over 800 million students, more than half the world’s student population, still faced meaningful disruptions to their education—ranging from full school closures in 31 countries to reduced or part-time academic schedules in another 48 countries [1]. In Spain, high schools were shut down in March 2020 (moving their curriculum fully online) and reopened with social distancing measures and blended learning in September 2020.

Evidence to support the effectiveness of global school closures in controlling COVID-19 is sparse, but the harms related to prolonged school closure are well documented [2]. Therefore, many governments decided to reopen their schools in different phases, introducing social distancing and testing and tracing regimes. Furthermore, instructional models became more flexible in search of quality of education, while guaranteeing adequate security measures [3]. These models included online learning, which has reached unprecedented levels, and blended learning, where students rotate between online and traditional content on fixed schedules, allowing that every student need not come into physical classrooms in face-to-face schools in the same space at the same time. The role of blended learning as a key factor to keep education running has been heightened exponentially during the COVID-19 pandemic [4].

The advent of COVID-19 has caused many transformations in all educational subjects —especially in those subjects, such as physical education (PE) in secondary school, which have been traditionally considered a practical subject, where close proximity and physical contact is common [5]. Besides previous concerns regarding the use of digital technologies in PE [6,7], PE teachers had to seek to manage an important tension between the experiential nature of PE as a subject, and the institutional and external constraints towards online and blended approaches [8]. As Daum and Buschner [9] indicated, PE is physical by nature, and remote instruction seems counterintuitive. Within this new framework, the huge changes in the delivery of PE have brought significant consequences for PE teachers, who have been tasked with making adaptations to their traditional teaching practices to deliver quality educational experiences—dealing with unique challenges such as the teaching and learning of motor and sport skills, dance or fitness [10]. Furthermore, new responsibilities for PE teachers arise, considering the role that PE could have in responding to the immediate physical and mental health effects of the current health pandemic [11].

Prior systematic review studies [9,12,13] on online and blended learning in PE showed that the research regarding blended learning in high school PE is very limited, and somewhat disconnected. Some studies have analyzed high school students’ and teachers’ perceptions of a hybrid PE course, but each study included small samples, with only one teacher participant and in a context prior to the COVID-19 pandemic [14,15]. 

It is relevant to point out that the blended or online learning defined in the literature up to 2019 [12] and the current distance learning experience due to the pandemic have some significant differences. Before the eruption of the COVID-19 pandemic, online and blended PE were a matter of choice for teachers and students. fitting to their needs. During the pandemic setting, however, online and blended learning have become a necessary mode of instruction in schools, regardless of the preferences of teachers and students [10].

In COVID-19 times, most research about blended learning PE has been performed within higher education, e.g., analyzing the impact of blended learning in PE teacher education programs [8,16]. Nevertheless, several studies have analyzed the way that elementary and high school PE teachers dealt with the fully online learning PE experience during the quarantine caused by COVID-19 in 2020; these studies described the changes in teaching interventions and included recommendations and specific proposals to teach PE when facing closed schools and distance learning [17,18,19].

Likewise, there are a few studies focused on the role of PE teachers during the initial phase of the COVID-19 pandemic in the United States. For instance, one investigation explored the types of support PE teachers—teaching in a variety of grade levels—need, and their concerns about a continued shift toward distance learning [20]. Recently, a survey study was carried out to investigate elementary and secondary PE teachers’ experiences, perceptions, and needs when implementing online PE, although they did not perform a differentiated analysis on blended learning [21,22].

To the best of our knowledge, no studies have analyzed the positive and negative feedback of high school PE teachers towards the shifting from remote learning during the first phase of the COVID-19 pandemic to blended learning one year after the initial outbreak of the COVID-19 pandemic. The understanding of educators’ experience during the pandemic might help blended learning be carried out better in the future, taking advantage of the experience to improve PE in usual scenarios other than the pandemic emergency. The more we know about how PE teachers faced the blended learning experience during the COVID-19 pandemic, the better to guide both present and future policies and procedures for blended PE. Thus, the current research explores the high school PE teachers’ perceptions about the potential, advantages, and disadvantages of the blended learning teaching model one year after the onset of the COVID-19 pandemic.

## 2. Materials and Methods

### 2.1. Participants

A sample of 174 high school PE teachers from Spain (120 men and 54 women) participated in this research. Regarding age, 12 teachers self-reported an age under 30 years, 62 teachers between 30 and 39 years, 72 teachers between 40 and 49 years, and 28 teachers were more than 50 years old. Concerning teaching experience, 46 teachers indicated a teaching experience of less than 5 years, 13 teachers had a teaching experience ranging from 5 and 9 years, 21 teachers had a teaching experience from 10 to 14 years, 34 teachers from 15 to 19 years, 37 teachers from 20 to 24 years, and 10 teachers from 25 to 29 years, while 13 teachers had a teaching experience of over 30 years. Concerning educational levels, 100 teachers self-reported exclusively giving classes in compulsory secondary education, 27 teachers only gave classes in post-compulsory secondary education, while 47 teachers combined compulsory and post-compulsory secondary education. In relation to socioeconomic status, six teachers self-reported that the socioeconomic level of the students’ families was low, 64 teachers indicated a low middle socioeconomic level, 95 teachers a middle socioeconomic level, while 9 teachers self-reported a high socioeconomic level. The teachers participating in this study worked mainly in public schools (151 compared to 23 who taught in private schools). The participants were recruited and selected following a purposive sampling method. 

### 2.2. Measures

#### 2.2.1. Teachers’ Perceptions of Blended Learning

To examine teachers’ perceptions towards blended learning teaching models, a survey was developed and administered. The researcher team provided a first panel of three experts on PE teaching (i.e., a PE teacher and two university professors) with information to elaborate the questions and responses for the survey. Specifically, the experts developed the questionnaire based on the Royal Decree-Law 1105/2014 [23] (Ministerio de Educación, 2015) for curricular contents and assessment in secondary school PE, and previous research such as Delgado's [24] spectrum of teaching styles (broadly known among Spanish PE teachers), articles by Fernandez-Rio et al. [25,26] for instruction, Buschner’s [27] proposal of online PE advantages and disadvantages, and Daum and Buschner’s [9] review on online and blended learning PE. In turn, ten didactic programs of secondary education PE, from the 2018/2019 academic year, from different educational centers in Spain were analyzed, in order to select the curricular aspects and their organization prior to COVID-19. Similarly, questions were included about the use of new methodological strategies such as flipped learning, gamification, or challenge-based learning [28]. Based on this information, the experts elaborated three items for curricular contents, four items for instruction, and two items for assessment. They also decided unanimously to include several types of response choices, considering multiple selection, Likert-type scales, and open-ended responses.

A second panel of four experts on PE teaching (i.e., two PE teachers and two university professors) independently and qualitatively judged the content of every survey item. This panel made an assessment considering the curricular elements and programs that were taught in the fully face-to-face teaching of PE prior to COVID-19. For this qualitative analysis, each expert assessed the clarity and understanding of each item through a 5-point-likert type scale [29]. Next, Aiken’s V index [30] was computed to gather content validity evidence for the survey. This coefficient is suitable when the lower limit of its 95% confidence interval (95% CI) is equal to 0.70 or greater [31]. Although acceptable Aiken’s V scores were obtained for clarity and understanding, the experts proposed slight modifications for two items. The main researcher considered these suggestions, implying the redrafting of those two items. The survey’s new version was qualitatively analyzed by four new experts (i.e., two PE teachers and two university professors). Again, satisfactory Aiken’s V values were found and no proposals were received. Finally, a pilot study was developed with two PE teachers, who checked the correct understanding and clarity of each survey item and responses. The final questionnaire included multiple-choice questions and Likert-type scales in which participants specified their level of agreement to a statement in five points: (1) Strongly disagree; (2) Disagree; (3) Neither agree nor disagree; (4) Agree; (5) Strongly agree.

#### 2.2.2. Sociodemographic Variables

For sociodemographic variables, a questionnaire collecting the teachers’ age, their teaching experience, the educational level where they gave classes, and the socioeconomic level of the students’ families was used. 

This manuscript focuses on data from the questions related to teachers’ perceptions about the advantages and disadvantages of the blended learning teaching model, and the data related to contents, instruction, and assessment will be published separately. 

### 2.3. Procedure

An integrated web-based application (Google Forms) was used to administer the online survey to Spanish high school teachers from January to February of the 2020/2021 school year. The survey was sent to corporate schools’ emails and the professional association of PE graduates of Spain. As a previous step, the survey included informed consent to be compulsorily provided by the potential participants. The following step included information and instructions to fill in and pointed out that participation was fully voluntary and anonymous. The guidelines also reported that there were no right or wrong answers given. and that the researchers only aimed to learn teachers’ perceptions of blended learning. Data were confidential and treated with exclusively academic and research goals in accordance with the ethical standards for research on human beings proposed in the Helsinki Declaration. The average time for completion was approximately 10 minutes.

### 2.4. Data Analysis

Data were analyzed using IBM SPSS (version 23.00). Before the main analyses, the data were screened to detect potential univariate outliers (i.e., *Z* scores over 3) and multivariate outliers (i.e., Mahalanobis d^2^ at *p* < 0.001) [32]. A total of seven cases were identified as univariate outliers and two as multivariate outliers, which were removed. The final sample was of 174 secondary PE teachers for the remaining analyses. The normality assumption was assessed by skewness and kurtosis coefficients, showing that standardized values up to 1.96 would underpin a normal data distribution [32]. To inform descriptive statistics, mean scores together with standard deviation and relative frequencies were, respectively, computed for each variable under study. Gender differences were analyzed by independent t-tests, while analysis of variance (ANOVA) tests were used to examine differences by age, teaching experiences, educational level and socioeconomic status. In those cases, in which statistically significant differences were found, the Bonferroni correction was applied to determine between which groups this difference existed. For categorial variables, differences by gender, age, teaching experiences, educational level and socioeconomic status were examined using Pearson’s chi-squared (χ^2^) tests. Complementary to the level of statistical significance (*p* < 0.05), effect size was estimated by Cohen’s d measure for independent t-test, partial eta squared (ƞ_p_^2^) for ANOVA tests, and Cramer’s V for χ^2^ tests. Effect sizes are shown to be insignificant with values up to 0.10, small with values between 0.10 and 0.30, medium with values between 0.31 and 0.50, and large with values over 0.50, respectively [32].

## 3. Results

The results of the assessment of normality assumption are shown in Table 1. Specifically, there were standardized scores from −1.31 to 0.97 for skewness and from −1.03 to 1.12 for kurtosis, which gathered evidence in support of the normality assumption. 

On the other hand, Table 1 also displays the teachers’ perceptions of the PE blended teaching–learning model. In particular, the high school PE teachers obtained higher mean scores than the mid-point of the measurement scale (4.14 out of 5) with regard to a greater dedication given with the blended learning model than the face-to-face one. Indeed, 127 (72.99%) teachers showed their agreement with a higher dedication, while 31 (17.82%) indicated a perception of neither agreement nor disagreement, and 16 (9.20%) teachers reported their disagreement. Regarding teamwork and coordination with other teachers, teachers scored over the mid-point of the 5-point measurement scale. Specifically, 83 (47.70%) teachers agreed or strongly agreed that teamwork and coordination with other teachers may have helped teachers to deal with the challenge of blended teaching. In fact, 42 (21.14%) teachers positioned themselves as neither agreeing nor disagreeing, while 49 (28.16%) teachers disagreed with this point. Concerning the improvement of students’ autonomy involved in this type of instructional model, teachers scored slightly above the mid-point of the measurement scale. Specifically, 76 (43.68%) teachers showed disagreement or strongly disagreement with the gain of students’ autonomy, while 55 (31.61%) teachers positioned themselves as neither agreeing nor disagreeing, and only 43 (24.71%) teachers showed agreement or strong agreement with a higher autonomy exhibited by students.

Referring to the enhancement in the students’ use of ITC, PE teachers scored higher than the mid-point of the measurement scale. Specifically, 106 (60.92%) PE teachers displayed agreement or strong agreement with the fact that blended learning improves students' use of new technologies, while 40 (23.99%) PE teachers reported neither agreement nor disagreement, and 28 (16.09%) PE teachers showed disagreement or strong disagreement. With regard to the increase of students’ motivation, PE teachers reported scores below the mid-point of the measurement scale. Specifically, 124 (71.26%) PE teachers disagreed or strongly disagreed with the fact the blended learning model increased the students’ level of motivation, while 37 (21.26%) PE teacher reported neither agreement nor disagreement and, conversely, 13 (7.47%) PE teachers displayed agreement or strong agreement. In reference to family involvement in the teaching–learning process under a blended instructional model, PE teachers scored on the mid-point of the measurement scale. Specifically, 90 (51.72%) PE teachers showed disagreement or strong disagreement, 49 (28.16%) teachers displayed neither agreement nor disagreement, and 35 (20.11%) PE teachers reported agreement or strong agreement. Regarding the lack of technological sources in students as a limitation, PE teachers reported a higher score than the mid-point of the measurement scale. In fact, 103 (59.20%) PE teachers agreed or strongly agreed with this limitation for students, while 41 (23.56%) PE teachers showed neither agreement nor disagreement, and 30 (17.24%) PE teachers disagreed or strongly disagreed with the lack of technological sources as a limitation.

Concerning the lack of knowledge in the use of ICT by students as a limitation, PE teachers obtained greater scores than the mid-point of the measurement scale. More specifically, 95 (54.98%) PE teachers agreed or strongly agreed with this limitation, 49 (28.16%) PE teachers positioned themselves as neither agreeing nor disagreeing, and 30 (17.24%) PE teachers disagreed or strongly disagreed that lack of knowledge in the use of ICT was a limitation. As far as the difficulty in establishing a relationship of trust with the student, PE teachers scored above the mid-point of the measurement scale. Indeed, 137 (78.74%) PE teachers reported agreement or strong agreement, 18 (10.34%) PE teachers displayed neither agreement nor disagreement, and 19 (10.92%) PE teachers showed disagreement or strong disagreement.

Table 2 shows non-significant differences by gender and age for each of the study variables. Similarly, Table 3 displays the absence of significant differences according to teaching experience, educational level, and socioeconomic status of the student’s families. However, there were differences regarding the difficulty in establishing relationships of trust with students (statement 10). Specifically, in comparison to teachers working in upper class students’ schools, teachers working in middle- and low-class students’ schools found it much more difficult to establish, with blended learning, a socio-affective relationship between students (*p* = 0.003), and between teachers and students (*p* = 0.025; Figure 1 and Figure 2). In addition, significant differences were found in the lack of technological means of the students as a limitation (statement 9), regarding the socioeconomic status of the students’ families. Particularly, teachers who work in middle- and low-class students’ schools considered both the lack of technological means of the students and the lack of knowledge in the use of information and communication technologies (ICT) by students as a limitation to a greater extent than teachers working in upper class students’ schools (*p* = 0.022 and *p* = 0.038, respectively; Figure 3 and Figure 4).

In relation to PE teachers’ perceptions of students’ levels of physical activity, 155 (89.08%) had the perception that students obtained lower levels of physical activity during a blended-learning model, while only 2 (1.15%) teachers perceived higher levels of physical activity, and 17 (9.77%) held the perception that students had the same levels of physical activity in a blended-learning model as in a face-to-face model. This perception was invariant across gender (χ^2^[df = 2] = 2.43, *p* = 0.297, V = 0.11), age (χ^2^[df = 6] = 6.83, *p* = 0.337, V = 0.14), teaching experience (χ^2^[df = 9] = 9.88, *p* = 0.626, V = 0.16), educational level (χ^2^[df = 4] = 2.04, *p* = 0.731, V = 0.08), and socioeconomic status (χ^2^[df = 6] = 8.97, *p* = 0.175, V = 0.19). 

## 4. Discussion

The purpose of this article was to examine PE teacher perceptions on the advantages and disadvantages of blended learning in relation to traditional face-to-face learning. The main findings showed the challenges PE teachers are facing when implementing blended learning and the issues derived from the digital dependence of this mode of PE delivery.

The PE teachers participating in this study believed that blended learning implied extra work compared to fully face-to-face instruction. Previous published hybrid PE experiences agree with this perception, as teachers reported that the use of online tools for teaching and the added online interaction with students were time-demanding and resulted in increased out-of-class workload [15,33,34]. During the first wave of the COVID-19 pandemic, a group of students in initial training to become teachers expressed their concerns about the increased probability of suffering from teacher stress when following an online teaching methodology [35]. To avoid getting overwhelmed with online and blended learning, Killian et al. [36] suggest that it is necessary to balance, in a manageable way, training, planning, synchronous class meetings, office hours and other teaching responsibilities.

There are many elements that may influence this workload perception, such as the fact that many PE teachers felt unprepared to use technology before the pandemic [37], or the lack of administrative support for additional training to become more effective remote instructors [22]. On the other hand, teamwork has the potential to reduce workload, and social support is an important tool for navigating work overload [38]. Around half of the PE teachers surveyed in our study agreed that teamwork and coordination with other teachers may have helped teachers to deal with the challenge of blended teaching, but further investigation is needed to analyze the relation between collaboration with peers and workload in online and blended settings.

Nevertheless, in general, it seems that teachers have learned a lot more about online technologies in PE than they ever had before, and this workload may turn into a training investment that should have positive implications in the future [21]. Likewise, virtual schooling can also develop students’ digital skills, which will be useful to them as they progress to the next stages of their life [39]. In fact, the PE teachers participating in the present study mostly agree that blended learning improves students’ ICT use. However, the PE teachers’ perceptions about the autonomy exhibited by students in the current study is mixed, and they neither agree nor disagree that blended learning improves students’ autonomy. Further research is needed to deepen the relation between PE blended learning and students’ autonomy. Although autonomous learning may be considered one of the strengths of online instruction, some PE teachers participating in previous studies viewed the inherent self-directed nature of online learning as a potential barrier, particularly for elementary students [12].

Regarding students’ motivation, the PE teachers surveyed in the current study reported that the use of blended learning did not enhance students’ motivation when compared to face-to-face teaching. This result questions one of the main potential advantages of online PE: that students are motivated by technology. Buschner [27] noted that implementing online physical education might have a beneficial motivational factor for students who had grown up using technology. Another potential advantage, according to Buschner [27], is that it fits students’ needs by using a personalized system of instruction. But the PE teachers in the present study do not indicate that blended learning allows a more individualized teaching–learning process.

Moreover, students in blended classrooms have also shown an increased motivation to learn [40,41], though the participants in these studies were university students, which could partly explain the difference with our study. Likewise, Østerlie [42] carried out an interventional study where secondary PE students participating in the blended learning intervention group increased their motivation to take part in PE, based on their expectancy beliefs and subjective task values for the subject, compared to students who participated in a traditional PE control group. The context in which Østerlie’s research was developed, in which student participation was voluntary and prior to the pandemic, could explain the differences in results with the present study.

In line with prior research that shows that family socioeconomic level and the availability of use of new technologies have effects on the school performance of students [43], the results of the current study suggest that the lower class endures a lack of technological means the most. As Van Lancker & Parolin [44] comment, students from lower-income families are likely to struggle to follow online courses because of their precarious housing situations. Unfortunately, all students do not have the same opportunities to connect online and have suitable digital devices and places in their homes. Even in a very rich developed country, such as the United States, PE teachers indicated that just half of their students had access to the technology required to effectively learn in a distance learning environment during the initial pandemic outbreak; moreover, rural PE teachers reported the least access for their students to technology and rated themselves as least effective in their remote PE teaching [22]. Likewise, another study revealed that equity issues are a prominent concern for PE teachers when performing distance learning during COVID times [20].

The PE teachers surveyed in our study pointed out that not only the lack of ICT, but also the lack of knowledge in the use of ICT by students was a limitation. This result brings to light the relevance of so-called digital illiteracy, referring to the fact that although people have access to new technologies, they do not know how to use them [45]. Besides access to technology, previous research shows that the students that are successful in online learning environments are those who have independent orientations towards learning, who are highly motivated by intrinsic sources, and who have good aptitude to manage their own time, literacy, and technology skills [39]. This set of characteristics may leave behind many high school students when implementing blended PE learning, regardless of technology issues.

The results of the present study seem to indicate that the use of the blended model does not necessarily facilitate greater family involvement than the face-to-face model. Daum et al. [10] state that parents, in the distance learning models, become a kind of gatekeeper, as they may need to help the student with technology and internet issues, offer support with assignments, assist with instructions, find materials, and other educational necessities. The issue is that the inability of some parents to supervise their children’s distance learning activities reflects the socio-cultural differences that the school aims to reduce, and thus remains a concern for teachers [20]. The specific recommendations proposed by Daum et al. [10] to meet the needs for low-income/disadvantaged students may help PE teachers to manage the complexities of distance learning in PE and ensure that their students are ready for blended learning.

It appears that blended and online learning can provide expanded educational access for students [39] but, at the same time, may represent a barrier for some of them, challenging educational equality, and widening the learning gap between children from lower-income and higher-income families [27]. This digital gap reopens a dilemma on distance learning and reinforces the perspective of those who think that virtual classes should not be the only option for core or elective subjects, but rather a supplemental option for students [46], tipping the scale in favor of the blended learning instead of fully online learning.

As for social relationships, the present study reveals that PE teachers consider that the blended model makes it more difficult to establish social relationships, both between teachers and students, and between students, compared to the face-to-face model. Previous PE blended learning research confirms the teachers’ perceptions identified in our study, showing how teachers miss the face-to-face interactions with the students, feel disconnected from the students, and express concerns related to a potential lack of students’ socio-relational learning opportunities in a virtual environment [14,15,34]. Taking classes online rather than in school causes many teachers and parents to worry that students will lose the socialization aspect of school PE classes [46]. Daum et al. [10] outline strategies aimed at fostering and promoting interactions between teachers and students and between students that may help teachers to deal with this issue. Furthermore, some studies present new possibilities for internet use in PE and demonstrate how the internet can connect students to others, i.e., participating in online exergaming or group exercise experiences together via online platforms [12,20]. It appears that teacher training in these interaction strategies may help to improve social relationships in non-contact contexts.

The perceptions of PE teachers in our study, considering that the physical activity performed by students during blended learning had been lower than usual, are in accordance with evidence that shows that one of the immediate consequences of the COVID-19 outbreak is that children and youths had lower physical activity levels and higher sedentary behavior [47,48]. The lost physical activity time is expected to have severe health effects [49,50]. This adverse impact on movement enhances the significance of quality PE experiences during the COVID-19 pandemic to promote students’ meaningful engagement in physical activity opportunities [36].

However, research prior to the pandemic highlights the role of blended PE in the promotion of physical activity, as it has the potential to minimize instruction time—improving the efficiency of instructional delivery, and to maximize physical activity—enhancing active learning opportunities [12]. In fact, compared to direct instruction, blended learning may expand instruction beyond the gym and allow for extended in-gym moderate to vigorous physical activity [51]. Likewise, previous studies have shown that the proper use of home-available and low-cost exergames and smartphones is a promising alternative for increasing physical activity in remote and blended learning PE [52,53,54].

The context drawn by the pandemic represents a significant departure from blended learning as defined in the literature and may explain this apparent controversy. Confinement at-home measures, the closures of gyms and public spaces, and physical distancing measures have limited students’ access to school and community physical activity environments, making remaining physically active difficult for youths [36,55].

This research presents some limitations. Since the survey included self-rated scale items, it is unknown what criteria teachers were using for measurements. Moreover, the results of this study are based on only teacher self-reporting and on the analysis of one discipline. Further research should consider other elements of the educational system, such as students, parents, and administrative staff, to explore other perspectives and to contrast and triangulate these findings. Likewise, the analysis of the perspectives of high-school teachers of other disciplines would allow the assessment of whether similarities exist between different disciplines. Another limitation is that the survey as a data collection method did not allow any further elucidation of the answers provided to better understand teachers’ opinions and perceptions. Future studies should include more interactive and in-depth data record instruments, such as interviews.

Finally, as in any other cross-sectional study, this research analyzes the data at a specific point of time (one year after the initial pandemic outbreak). Follow-up studies are needed to monitor the evolution of PE teachers’ perceptions during the course of the pandemic and afterwards. 

## 5. Conclusions

This research may contribute to understanding how PE teachers experienced blended learning one year after the beginning of the COVID-19 pandemic, as it sheds light on their perceptions about important and common questions related to this mode of instruction. Despite the fact that it is situation-specific, the authors believe that this study is relevant, and the results could help to inform the future of blended PE.

Although the blended model allows the overcoming of two weak points of the full online learning PE model, i.e., minimal student socialization and physical activity [9], these two issues continue to be a concern for PE teachers in the blended learning model when compared with traditional in-presence learning. It is suggested to opt for blended learning approaches that maximize face-to-face time and explore new strategies during online time to address these questions.

The socioeconomic status of the students may be a limitation for the expansion of blended learning. Given the potential threat of widening the learning divide between students from lower-income and higher-income families when using blended learning, high school PE teachers should adapt their learning resources for adolescents with no reliable internet connection, a computer, or adequate place to study. Furthermore, these adaptations should be accompanied by a significant investment in attempting to deal with the digital gap by providing devices and internet connections to students, and—just as important as the equipment—additional training on the proper use of the new technologies to maximize learning for those students who need it most.

Likewise, PE teachers need assistance to manage the work overload implied by the transition to blended learning. The need to provide initial and ongoing training on how to use PE blended learning and ICT is evident. Future teachers should be prepared for the possibility of teaching PE blended and online. Moreover, COVID-19 has forced physical educators to teach PE with this mode of PE delivery, due to convenience and not from freedom of choice, without solid research evidence to guide their practice.

The COVID-19 experience with blended education could bring about some positive changes for the PE profession, e.g., teachers more used to ICT. In any case, when the educational landscape imposed by COVID-19 shifts to the “new normality” scenario, live PE classes should not be reduced without enough evidence to validate the usefulness of blended education. In any case, based on the findings of the current study, before implementing blended learning in PE, we should ensure that it does not imply a risk of increasing inequality due to ICT access issues. Likewise, PE teachers should have support and training to guarantee that blended learning helps to optimize PE face-to-face time and enhance social relationships and physical activity using online resources, e.g., exergames and smartphones apps. Further studies should explore the potential, expand understanding, and determine the best practices and the efficacy of using blended learning in PE. 

## Figures and Tables

**Figure 1 ijerph-18-11146-f001:**
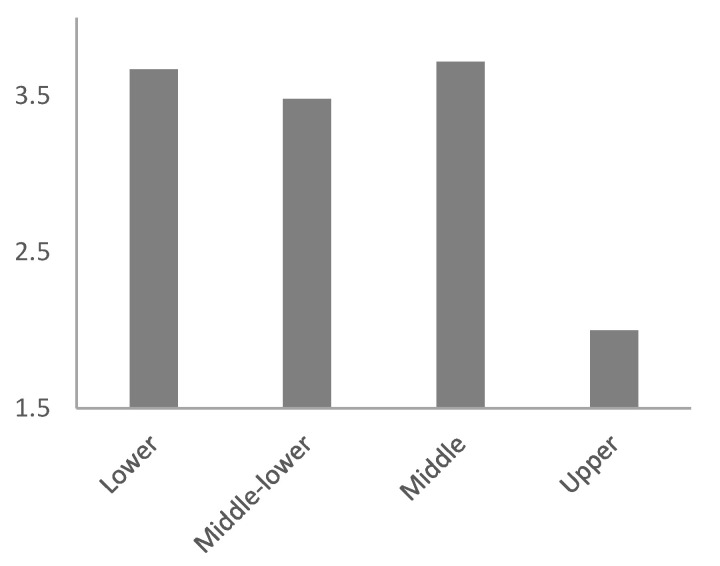
Average degree of teachers’ agreement with the statement "It is more difficult to establish a relationship of trust with the student", according to the socio-economic status of students.

**Figure 2 ijerph-18-11146-f002:**
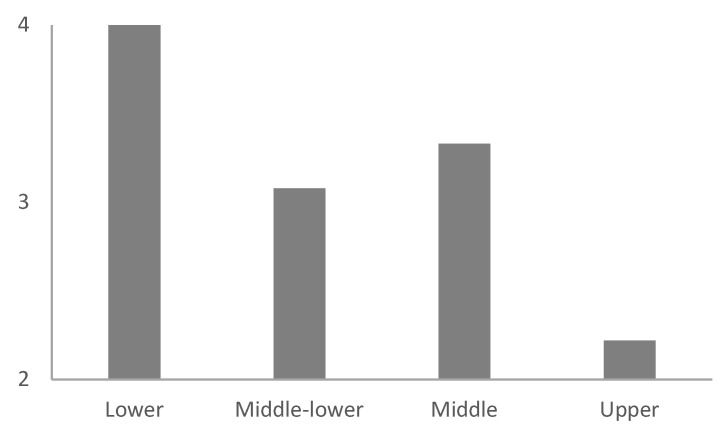
Average degree of teachers’ agreement with the statement "Socio-affective relationships among students have worsened", according to the socio-economic status of students.

**Figure 3 ijerph-18-11146-f003:**
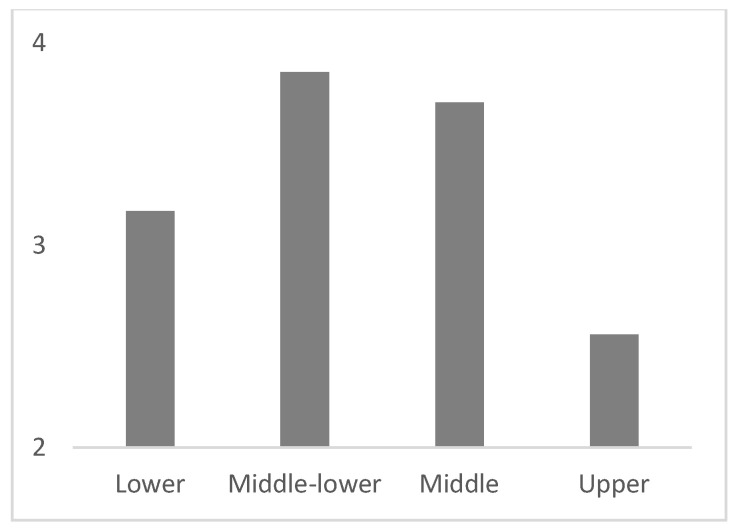
Average degree of teachers’ agreement with the statement "The lack of technological means in the students is a limitation", according to the socio-economic status of students.

**Figure 4 ijerph-18-11146-f004:**
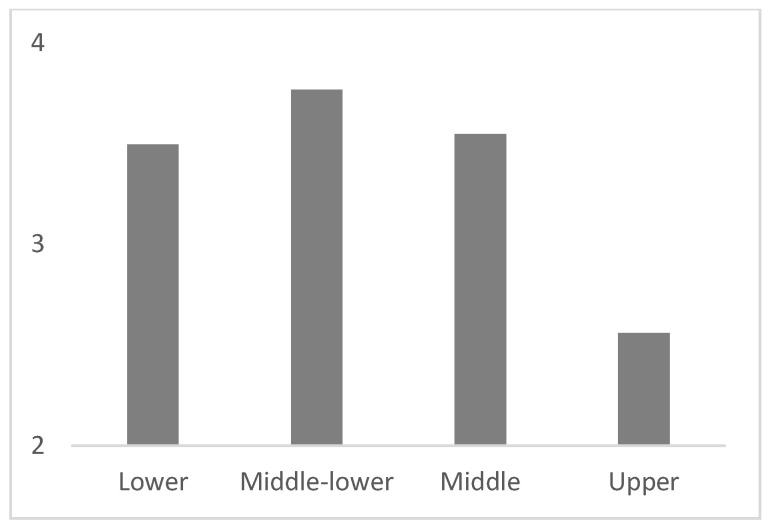
Average degree of teachers’ agreement with the statement "The lack of knowledge in the use of information and communication technologies by students is a limitation", according to the socio-economic status of students.

**Table 1 ijerph-18-11146-t001:** Average degree of teachers’ agreement with the following statements, regarding teaching–learning in the blended model compared to fully face-to-face model (1 = Strongly disagree; 5 = Strongly agree).

Statement	M	SD	γ_1_	γ_2_
1. It has meant more dedication than the face-to-face	4.14	1.16	−1.24	0.67
2. Teamwork and the coordination with other teachers have helped me	3.33	1.30	−0.26	−1.03
3. Improves student’s autonomy	2.72	1.17	0.22	−0.71
4. Improves the students’ use of new technologies	3.66	1.14	−0.62	−0.34
5. Increases student’s motivation	1.99	1.05	0.97	0.49
6. Allows a more individualized teaching−learning process	2.48	1.19	0.35	−0.76
7. Facilitates involving families in the teaching−learning process	2.50	1.23	0.43	−0.72
8. The lack of technological means in the students is a limitation	3.68	1.26	−0.65	−0.55
9. The lack of knowledge in the use of information and communication technologies by students is a limitation	3.57	1.19	−1.31	1.12
10. It is more difficult to establish a relationship of trust with the student	4.04	1.12	−0.52	−0.90
11. Socio-affective relationships among students have worsened	3.20	1.27	−0.15	−0.81

**Table 2 ijerph-18-11146-t002:** Differences by teacher gender and age for the study variables.

Statement	Gender			Age		
	t (df = 172)	*p*-Value	d	F (df = 3)	*p*-Value	ƞ_p_^2^
1. It has meant more dedication than the face-to-face	0.81	0.421	0.14	0.41	0.742	0.01
2. Teamwork and the coordination with other teachers have helped me	0.08	0.936	0.02	2.31	0.078	0.04
3. Improves student’s autonomy	0.94	0.351	0.14	1.20	0.313	0.03
4. Improves the students’ use of new technologies	1.02	0.309	0.16	2.82	0.062	0.07
5. Increases student’s motivation	0.10	0.924	0.02	0.84	0.475	0.02
6. Allows a more individualized teaching-learning process	1.07	0.285	0.16	1.05	0.376	0.03
7. Facilitates involving families in the teaching-learning process	1.14	0.256	0.17	0.43	0.731	0.01
8. The lack of technological means in the students is a limitation	1.06	0.311	0.16	1.80	0.148	0.05
9. The lack of knowledge in the use of information and communication technologies by students is a limitation	1.05	0.298	0.16	0.95	0.420	0.03
10. It is more difficult to establish a relationship of trust with the student	0.57	0.571	0.09	0.90	0.446	0.02
11. Socio-affective relationships among students have worsened	1.38	0.168	0.16	0.78	0.509	0.02

**Table 3 ijerph-18-11146-t003:** Differences by teaching experiences, educational level, and socioeconomic status for the study variables.

	Teaching Experience			Educational Level			Socioeconomic Status		
Statement	F (df = 6)	*p*-Value	ƞ_p_^2^	F (df = 2)	*p*-Value	ƞ_p_^2^	F (df = 3)	*p*-Value	ƞ_p_^2^
1. It has meant more dedication than the face-to-face	0.81	0.567	0.03	0.93	0.397	0.01	4.41	0.059	0.08
2. Teamwork and the coordination with other teachers have helped me	1.96	0.076	0.07	1.19	0.397	0.01	1.88	0.135	0.03
3. Improves student’s autonomy	0.55	0.768	0.03	0.23	0.779	0.01	1.00	0.398	0.03
4. Improves the students’ use of new technologies	0.54	0.773	0.03	2.14	0.122	0.04	0.13	0.941	0.01
5. Increases student’s motivation	1.14	0.346	0.06	1.98	0.143	0.03	0.88	0.456	0.02
6. Allows a more individualized teaching-learning process	1.45	0.202	0.07	1.40	0.251	0.02	1.90	0.134	0.05
7. Facilitates involving families in the teaching-learning process	0.52	0.792	0.03	2.17	0.119	0.04	0.23	0.162	0.01
8. The lack of technological means in the students is a limitation	1.44	0.206	0.07	1.02	0.363	0.03	2.74	0.044	0.07
9. The lack of knowledge in the use of information and communication technologies by students is a limitation	1.16	0.336	0.06	0.15	0.859	0.01	1.21	0.309	0.03
10. It is more difficult to establish a relationship of trust with the student	0.88	0.416	0.02	2.81	0.014	0.13	2.12	0.102	0.05
11. Socio-affective relationships among students have worsened	1.07	0.383	0.06	1.55	0.214	0.02	1.28	0.286	0.03

## Data Availability

The data presented in this study are available on request from the corresponding author. The data are not publicly available due to privacy regulations.

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
