# Peer review of "High School Physical Education Teachers’ Perceptions of Blended Learning One Year after the Onset of the COVID-19 Pandemic"

_ijerph, 2021, doi:10.3390/ijerph182111146_

Round 1

Reviewer 1 Report

In general the content of the manuscript is fine. I would suggest some editing for language and grammar, however. Some areas were worded a bit strangely; not enough to obscure the intent, but it could be smoother. I also wonder about the statement in the abstract (as well as the text) saying that blended learning did not increase students' motivation. Was it expected to? It also seems like a nice way to wrap this up would be a short bit about whether/how/should blended learning continue to be used for PE and if so, under what circumstances.

Author Response

We want to thank the reviewer for investing his/her time in improving our manuscript. We have highlighted the changes made in the updated version of the manuscript by using red-colored text (see attached document). Below, we provide the point-by-point responses to the very thoughtful comments carried out by the reviewer. Thank you very much indeed. In this document, we indicate the modifications done in the new version of the manuscript, in addition to specifying, where appropriate, the page(s) where such modifications can be found.

Point 1: In general, the content of the manuscript is fine. I would suggest some editing for language and grammar, however. Some areas were worded a bit strangely; not enough to obscure the intent, but it could be smoother.

Response 1: Thank you for the comment. We have reviewed the manuscript to improve the wording, mainly the materials and methods section.

Point 2: I also wonder about the statement in the abstract (as well as the text) saying that blended learning did not increase students' motivation. Was it expected to?

Response 2: Previous studies had shown that online and blended learning might have a beneficial motivational factor in physical education secondary students [1,2], so it was expected that blended learning could increase students' motivation. However, the physical education teachers surveyed in the current study reported that the use of blended learning did not enhance students’ motivation when comparing to face-to-face teaching. As we comment in the discussion section, the context in which previous research was developed (i.e., participation in blended learning was voluntary and prior to the pandemic), could explain the differences in results with the present study.

Point 3: It also seems like a nice way to wrap this up would be a short bit about whether/how/should blended learning continue to be used for PE and if so, under what circumstances.

Response 3: In accordance with the reviewer’s comment, we have highlighted in the conclusions section the circumstances under which blended learning should continue to be used for PE (page 12, lines 477-482).

Reviewer 2 Report

The article analyzes the advantages and disadvantages perceived by Physical Education teachers in High Schools in relation to distance learning blended with face-to-face teaching.
In order to do so, a study is carried out with volunteers from this specific group of teachers.
The introduction provides a set of relevant and updated references on the subject, thus generating enough support to contextualize the study scientifically.
In materials and methods, a statistical description of the sample on which the study is carried out should be included, which will provide the basis for the statistical tests carried out in the following sections.
The procedure for the elaboration of the questionnaire and data preprocessing that has been carried out is sufficiently explained.
The socioeconomic variables used are adequate, although it would be convenient to have a statistical description of them.
As for the results obtained, the detail used is correct. However, it is difficult to have an overall view, as the measures are embedded in the text. It would make it easier to follow and understand the value of the results obtained if comprehensive tables of all the statistical tests carried out were drawn up. This would make it possible to relieve the text of quantitative interruptions and refer to the corresponding table.
The discussion is adequate and sufficient and the conclusions obtained are adequately supported by the results.
Finally, emphasis should be placed, both in the abstract and at the end of the introduction, on the contributions made in the article and the scientific gaps they fill.

An Appendix with the data or link to an open repository of your university is desirable in order to be able to reproduce the study.

Author Response

We want to thank the reviewer for investing his/her time in improving our manuscript. We have highlighted the changes made in the updated version of the manuscript by using red-colored text (see attached document). Below, we provide the point-by-point responses to the very thoughtful comments carried out by the reviewer. Thank you very much indeed. In this document, we indicate the modifications done in the new version of the manuscript, in addition to specifying, where appropriate, the page(s) where such modifications can be found.

Point 1: The article analyzes the advantages and disadvantages perceived by Physical Education teachers in High Schools in relation to distance learning blended with face-to-face teaching. In order to do so, a study is carried out with volunteers from this specific group of teachers. The introduction provides a set of relevant and updated references on the subject, thus generating enough support to contextualize the study scientifically.

Response 1: Thank you so much for your nice point.

Point 2: In materials and methods, a statistical description of the sample on which the study is carried out should be included, which will provide the basis for the statistical tests carried out in the following sections. The procedure for the elaboration of the questionnaire and data preprocessing that has been carried out is sufficiently explained. The socioeconomic variables used are adequate, although it would be convenient to have a statistical description of them.

Response 2: Upon the reviewer’s request, we have included additional information for a better description of the participating sample in terms of age, teaching experience, educational level, and socioeconomic status (page 3, lines 105-120).

Point 3. As for the results obtained, the detail used is correct. However, it is difficult to have an overall view, as the measures are embedded in the text. It would make it easier to follow and understand the value of the results obtained if comprehensive tables of all the statistical tests carried out were drawn up. This would make it possible to relieve the text of quantitative interruptions and refer to the corresponding table.

Response 3: In line with the reviewer’s comment, we have included two tables (table 2 and table 3) to highlight the obtained results with respect to the different analyses run. Likewise, we have redrafted the results section to present the findings in a brief and accurate manner in conjunction with the new tables.

Point 4: The discussion is adequate and sufficient and the conclusions obtained are adequately supported by the results. Finally, emphasis should be placed, both in the abstract and at the end of the introduction, on the contributions made in the article and the scientific gaps they fill.

Response 4: Thank you for the comment. Both in the abstract and at the end of the introduction, we have included new sentences that highlight the scientific gaps the article aims to fill, and the contributions made (pages 1-2, lines 16, 25-26, 94-97).

Point 5: An Appendix with the data or link to an open repository of your university is desirable in order to be able to reproduce the study.

Response 5: In accordance with the reviewer’s comment, the data used for this research has been uploaded to the platform and they are available on request from the corresponding author, as indicated in the “Data Availability Statement” section.

Reviewer 3 Report

The paper is well structured and comprehensive of the all the most relevant considerations about issues related to pandemic in education in general and in PE teaching in high school in particular. The aim is original. The experimental approach is well designed, and the statistical analysis is well conducted.

line 50: authors should specify the educational stage they are referring to. Whether they are referring to primary or school PE teachers or to both. This is very important, since the same may apply to higher education but the authors should limit their focus on a well-defined education stage.

line 111: it is not clear to me whether the questionnaire was validated or not. If not, it’s fine but the authors should discuss the reasons why they chose a not-validate questionnaire in the discussion section as a limitation of the study. Is the questionnaire that depicted in Table 1?

line 153: a time period in months would be better here.

line 158: it is not clear how participants obtained and gave the informed consent (also via google form?)

Figures: I would suggest including Figure 1, 2, 3 and 4 in a single 2x2 graph (with 4 sub-graphs).

discussion section: I understand that there is little research on this topic focused on high-school PE teachers, but it would be nice if the authors may possibly discuss the results (even part of them) of the present study in the light of similar research focused on satisfaction’s levels of high-school teachers of other disciplines in blended learning to assess whether similarities exist between different disciplines.

line 414-415: please consider that an alternative to PE remote and blended learning may be found in the emerging use of home-available (and low-cost) exergames and smartphones (https://doi.org/10.1007/978-3-030-31284-8_17). I think the authors should briefly discus about this, also in contrast to what stated by Daum and Buschner that indicated PE remote delivery teaching as counterintuitive.

Author Response

We want to thank the reviewer for investing his/her time in improving our manuscript. We have highlighted the changes made in the updated version of the manuscript by using red-colored text (see attached document). Below, we provide the point-by-point responses to the very thoughtful comments carried out by the reviewer. Thank you very much indeed. In this document, we indicate the modifications done in the new version of the manuscript, in addition to specifying, where appropriate, the page(s) where such modifications can be found.

Point 1: The paper is well structured and comprehensive of the all the most relevant considerations about issues related to pandemic in education in general and in PE teaching in high school in particular. The aim is original. The experimental approach is well designed, and the statistical analysis is well conducted.

Response 1: We would like to thank you for your nice words.

Point 2: Line 50: authors should specify the educational stage they are referring to. Whether they are referring to primary or school PE teachers or to both. This is very important, since the same may apply to higher education but the authors should limit their focus on a well-defined education stage.

Response 2: In line with the comment proposed by the reviewer, we have specified that PE refers to secondary school (page 2, line 52).

Point 3: Line 111: it is not clear to me whether the questionnaire was validated or not. If not, it’s fine but the authors should discuss the reasons why they chose a not-validate questionnaire in the discussion section as a limitation of the study. Is the questionnaire that depicted in Table 1?

Response 3: With respect to the reviewer’s comment, we would like to consider that, to the best of our knowledge, the survey followed a validation process by providing validity evidence based on the instrument’s content [1]. In particular, this evidence was gathered by two expert judgments, a pilot study, and the estimation of content validity measures (page 3, lines 123-149; page 4, lines 150-158). Given that survey consists of 1 item to measure every variable, psychometric analyses cannot be run to provide additional validity and reliability evidence by being required at least three items per variable for these tests [2,3]. Taking into account this rationale, we consider that the survey used for data collection is valid to measure every variable under study.

On the other hand, we should indicate that the items of the survey are effectively shown in Table 1.

Point 4. Line 153: A time period in months would be better here.

Response 4: Following the suggestion made by the reviewer, we have replaced “end of the first trimester” with “from January to February” (page 4, lines 168).

Point 5: Line 158: It is not clear how participants obtained and gave the informed consent (also via google form?)

Response 5: Thank you so much for this consideration. We have detailed that the informed consent was provided by the potential participants via google form, as a previous step to access the full survey (page 4, lines 170-171).

Point 6: Figures: I would suggest including Figure 1, 2, 3 and 4 in a single 2x2 graph (with 4 sub-graphs).

 Response 6: Thank you so much for this suggestion, which has been taken into consideration (page 8).

Point 7: Discussion section: I understand that there is little research on this topic focused on high-school PE teachers, but it would be nice if the authors may possibly discuss the results (even part of them) of the present study in the light of similar research focused on satisfaction’s levels of high-school teachers of other disciplines in blended learning to assess whether similarities exist between different disciplines.

Response 7: A comparative analysis where we could examine the way how high school teachers from different disciplines have dealt with the blended learning approach is interesting and necessary. Some studies show that teachers from other disciplines that require hands-on skills, as technology education, have reported closer difficulties to PE to teach in the emergency remote teaching [4], than other disciplines that do not require physical skills, as language or mathematics [5,6]. Nevertheless, we consider that this topic would deserve an in-depth analysis that exceeds the aims of the current study, so we have included it as a recommendation for future research. Thank you very much for the idea (page 11, lines 435-437).

Point 8: Line 414-415: please consider that an alternative to PE remote and blended learning may be found in the emerging use of home-available (and low-cost) exergames and smartphones (https://doi.org/10.1007/978-3-030-31284-8_17). I think the authors should briefly discus about this, also in contrast to what stated by Daum and Buschner that indicated PE remote delivery teaching as counterintuitive.

Response 8: We agree with the reviewer, and we thank the link to the document provided. In fact, we indicated in the discussion section that “some studies present new possibilities for internet use in PE and demonstrate how the internet can connect students to others, i.e., participating in online exergaming or group exercise experiences together via online platforms [12,20]. It appears that teacher training in these interaction strategies may help to improve social relationships in non-contact contexts.” Furthermore, in accordance with the reviewer’s comment, we have considered new references and included a sentence that point out the relevance of smartphones and exergames in PE:Likewise, previous studies have shown that the proper use of home-available (and low-cost) exergames and smartphones, is a promising alternative to increase physical activity in remote and blended learning PE [7–9]” (pages 11-12, lines 404-408, 477-482).

References

  1. Sireci, S.; Faulkner-Bond, M. Validity Evidence Based on Test Content. Psicothema 2014, 26, 100–107, doi:10.7334/psicothema2013.256.
  2. Hair, J.F.; Babin, B.J.; Anderson, R.E.; Black, W.C. Multivariate Data Analysis; 8th ed.; Cengage Learning EMEA: Andover, UK, 2018; ISBN 978-1-4737-5654-0.
  3. Kline, R.B. Principles and Practice of Structural Equation Modeling; 4th ed.; Guilford Press: New York, 2016; ISBN 978-1-4625-2334-4.
  4. Code, J.; Ralph, R.; Forde, K. Pandemic Designs for the Future: Perspectives of Technology Education Teachers during COVID-19. Information and Learning Sciences 2020, 121, 419–431, doi:10.1108/ILS-04-2020-0112.
  5. Almanthari, A.; Maulina, S.; Bruce, S. Secondary School Mathematics Teachers’ Views on e-Learning Implementation Barriers during the COVID-19 Pandemic: The Case of Indonesia. Eurasia Journal of Mathematics, Science and Technology Education 2020, 16, em1860.
  6. Lukas, B.A.; Yunus, M.M. ESL Teachers’ Challenges in Implementing E-Learning during COVID-19. International Journal of Learning, Teaching and Educational Research 2021, 20, 330–348.
  7. Gao, Z.; Chen, S.; Stodden, D.F. A Comparison of Children’s Physical Activity Levels in Physical Education, Recess, and Exergaming. Journal of Physical Activity and Health 2015, 12, 349–354, doi:10.1123/jpah.2013-0392.
  8. Kooiman, B.J.; Sheehan, D.P.; Wesolek, M.; Reategui, E. Exergaming for Physical Activity in Online Physical Education. International Journal of Distance Education Technologies (IJDET) 2016, 14, 1–16, doi:10.4018/IJDET.2016040101.
  9. Picerno, P.; Pecori, R.; Raviolo, P.; Ducange, P. Smartphones and Exergame Controllers as BYOD Solutions for the E-Tivities of an Online Sport and Exercise Sciences University Program. In Proceedings of the Higher Education Learning Methodologies and Technologies Online; Burgos, D., Cimitile, M., Ducange, P., Pecori, R., Picerno, P., Raviolo, P., Stracke, C.M., Eds.; Springer International Publishing: Cham, 2019; pp. 217–227.

Round 2

Reviewer 2 Report

The article has been considerably improved and has taken into account all the considerations suggested in the first revision.
A minor consideration is that the link to the RIUMA repository of the university could be included.